# Benchmarking Object-Centric Manipulation Using a Simulated Environment

May 31, 2020

- Original Paper : SAPIEN: A SimulAted Part-based Interactive ENvironment
- Paper written by : Fanbo Xiang, Yuzhe Qin, Kaichun Mo, Yikuan Xia, Hao Zhu, Fangchen Liu, Minghua Liu, Hanxiao Jiang, Yifu Yuan, He Wang, Li Yi, Angel Chang, Leonidas Guibas, Hao Su
- Retrospective written by : Hao Zhu

## Paper TL;DR

SAPIEN (SimulAted Part-based Interactive ENvironment) is a realistic and physics-rich simulated environment that hosts a large-scale set for articulated objects. It enables various robotic vision and interaction tasks that require detailed part-level understanding.

The core of SAPIEN components includes:

1. The engine, which provides physical simulation for articulated objects with pure Python interface and ROS integration;
2. The dataset, a collection of 2K articulated objects with joint annotations and rendering material;
3. The renderer, which renders scenes with OpenGL rasterizer and optionally Nvidia OptiX ray-tracer.

## Overall Outlook

Benchmark is of vital importance to evaluate the effectiveness of an algorithm. In recent years, the computer vision community benefits from using standard datasets and benchmarks. This trend is becoming more popular in the robotics community as well[1, 2, 3].

However, if we would like to bring the same comparing approach to the robotics community, there are some challenges we have to face:

- Robots and Robotics are complex, i.e. Robotics as a science has to analyze an entire autonomous system, not just one component, so the results in robotics papers are sometimes hard to replicate due to the hardware environment, which is impossible to be totally the same. This makes the decision on metrics and task descriptions harder.
- Benchmarking has to be closed-loop, i.e. the data changes with the action that is applied to the environment. We need to build an interactive environment and get feedback from the interaction. The equivalent in RL research would be on-policy learning.
- Quantitatively evaluating real robots may be difficult or even prohibitive due to difficulty in data collection and economic and security issues. Besides, the performance of a sensor-based robot is stochastic because each run of the robot is unrepeatable[4].

Is there an alternative way to benchmark the performance of robots? We think using a simulated environment is the way out. Compared to training robots by interacting with the real environment, using a simulated environment have irreplaceable advantages:

- Simulators allow quantitative and reproducible experiments, which enable the comparison of different algorithms on the same robot, environment and task. It is easy to replicate and benchmark an algorithm, as data is more convenient to generate and collect in the simulator.
- The experiment environment is easy to set up and the codes can be released to the community. So new researchers can get involved in this area without buying expensive hardware first.
- Simulators can help researchers estimate the distribution in algorithm performance and investigate the robustness of algorithm performance due to environmental factors.
- In most cases, training and testing the robot in the simulated environment is faster than in the real environment.

We considered using an existing environment at the very beginning, unfortunately, we found existing robotics simulators have different levels of simplification and focus, we have to make it ourselves to meet all the requirements.

Humans study the physical world from the interaction with objects, and we believe robots should study in the same way. As building home assistant robots is our final goal, we started from building a dataset of articulated objects for robots to interact with. We reunited members of the ShapeNet[5] crew and made ShapeNet movable. This dataset contains 14K movable parts over 2,346 3D articulated models from 46 common indoor object categories, richly annotated with kinematic pairs.

Finally, with the help of the SAPIEN robotics kit that supports URDF and ROS, researchers have the possibility to train and test the robot in the simulator and transfer it to the real world. However, simulators are not a faithful representation of reality. Whatever is learned in a simulator can typically not be applied in the real world. This problem requires further study.

# Opportunities for Improvement

## Flaws in the Dataset

Although we took a lot of effort building the dataset, it still has some problems:

- Segmentation related problems: As the first step of processing the 3D models, If the segmentation is too coarse, some movable parts are not separated, and if the segmentation is too detailed, there will be too many faces that we have to combine them into movable parts. But if we leave a single face behind, there will be a "flying piece". This mistake is easy to make since it is super frustrating to deal with more than hundreds of faces. Fortunately, we have checked many times to make sure the models we released are free from this problem.
- Lack of models in each category: The total number of objects in the dataset is large (more than 2K), but for each category, there are only ~100 models. Compared to some widely-used datasets in computer vision, we might need more data to support training and testing.
- Limited types of kinematic pair: SAPIEN supports prismatic joint and revolute joint. For the screw joint, we have to adopt a hardcoded way. Other types are not supported yet. However, there are more situations in the real world that we need to consider.

## More Tasks and Evaluation

Given the dataset of articulated objects, the protocols and benchmarks that define the experimental procedure and quantification methods are equally important. In the CVPR version, we present the benchmarks of vision algorithms for part detection and joint type recognition. We also show some interaction tasks that

SAPIEN supports by demonstrating heuristic approaches and reinforcement learning algorithms. These existing tasks are not really suitable for the environment. Obviously more robotic perception and interaction tasks with the protocol and benchmark are needed to define. I will explain more in the next section.

## Generalizability and Reality Gap

The simulator is a simplification of the real world, some details have to be simplified or eliminated, where the reality gap comes from, i.e. we have found the friction sometimes causes anomalies in the simulator. A common assumption is that trained models will be deployed under closed-set conditions. However, robots often have to deal with real world objects that were not covered by the dataset, which often requires robots to have the ability to handle unknown input data.

There are 3 levels of the generalizability of the object-centric manipulation:

- Training and testing using objects in the same category. E.g., training the robot to open the door, testing the robot to open a new door.
- Training and testing using objects in all the categories. E.g., training the robot to manipulate different kinds of objects, testing the robot using a new object from a known kind.
- Training using objects in some categories, testing with objects in a novel category. E.g., Training the robot to open doors and drawers, testing the robot with a refrigerator.

For the first level, we have shown the robot can successfully manipulate some objects in the simulator, but there are still some concerns: Can we apply the model trained from the simulated environment to the real robot? As could be expected, there will be a huge gap between the simulated and the real environment, so how to bridge the reality gap? From the easiest to the hardest level, How far can we go?
The answers to these questions still need a long time for us to explore.

# New Perspectives and Future Works

There is still a lot of work that can be done in the future:

1. **Dataset**
   Once we optimize the pipeline of building the dataset from 3D models, the system will be easier to use and available to the public. More accurate models with fine-grained annotations are needed to enlarge the dataset. More physical properties should be included.
   Since the paper was released, we have received some requests and questions about generating different types of data using SAPIEN. I understand researchers might not want to spend too much time dealing with the environment, but they still want to use the data to do some perception tasks, like pose estimation, state estimation, structure from motion, or so on. We will develop some easy-to-use tools to meet these requirements, including tools to generate images and videos from the objects -- although 3D models contain more information, it is more difficult to learn from the 3D models than from the 2D images or the videos.

2. **Environment**
   The first version of SAPIEN does not contain many scenes, we are building scenes with different layouts and objects to train robots in diversified scenarios.
   The dataset only contains articulated objects for now, but we also consider adding soft body and fluid simulation to make the environment possible for tasks like pouring water from a cup.

3. **Tasks and Evaluation**

Collins et al. [6] create a benchmark for simulated manipulation and evaluate the performance using object moving time, velocity, acceleration and Mahalanobis distance. Benchmarking object-centric manipulation tasks is much more complicated because of the change of states. For the articulated objects, how to define whether the robot successfully manipulates them? There are two strategies (from the perspective of state changes):

- By steps: The robot gets a score when it completes a step. For example, for the drawer, sometimes the robot does not open the drawer to the end, how much score should it gain? Successfully grabbing the handle (or other parts), moving in the correct direction, and the range of moving can be taken into account. Each step is given a weight to get a total score of a single trial.
- By results: We do not care about the steps, just set up a simple threshold. Take the door for example, if the robot can open the door for more than 90 degrees, we consider the robot makes it. However, the robot might not use the handle at some point.
These tasks and evaluation metrics are category-related, some objects may contain many kinematic pairs, so different strategies may apply to different categories or types, and the metrics have to be carefully designed.

# Summary

SAPIEN is a ready-to-use simulated environment for object-centric manipulation tasks, which can serve for both robotics and computer vision communities. We are still working on it to make it better and we hope more researchers get involved with us to explore more possibilities.

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
