# OpenReview forum: "Benchmarking Object-Centric Manipulation Using a Simulated Environment"
_roboticsfoundation.org/RSS/2020/Workshop/RobRetro — RobRetro 2020_

### Official Review · AnonReviewer1 · 2020-06-23
**Retrospective on really nice new simulation environment for the problem or manipulating and perceiving articulated objects. Would benefit from a deeper discussion in the context of Robotics Benchmarking.**

**Confidence:** 5
**Rating:** 9

**Review:**

The retrospectives “Benchmark Object-centric Manipulation Using a Simulated Environment” reviews the recent paper “SAPIEN: A SimulAted Part-based Interactive ENvironment” and adds some perspectives on this idea’s position within the landscape of robotics and computer vision benchmarks. Specifically valuable are the opportunities for improvement, new perspectives and future work. The environment itself is very interesting and a valuable addition to the robotics research landscape.

What I’m missing is a more substantial discussion of the difficulty of benchmarking in robotics and how it is more difficult than benchmarking in e.g. Computer Vision. Here are some points:
- Robots and Robotics are complex, i.e. Robotics as a science has to analyse an entire autonomous system not just one  component. This makes the decision on metrics and task descriptions harder.
- Benchmarking has to be closed-loop, i.e. the data changes with the action that is applied to the environment. The equivalent in RL research would be on-policy learning.
- Quantitatively evaluating real robots may be difficult or even prohibitive. Simulators allow quantitative and reproducible experiments. However, they are not a faithful representation of reality. Whatever is learned in a simulator can typically not be applied in the real world.

There may be more points to be made (see resources below), but the authors of this paper only refer to the last point on that simulations enable quantitative experiments and reproducibility even if you do not have a real robot. However, with their environment, they actually address the point on closed-loop benchmarking. Furthermore, the authors focus on the ‘irreplaceable advantages’ of simulators (which are correctly pointed out), but should also discuss the limitations of simulation in general that are mentioned a little bit on the last page in terms of reality gap.

The authors also mention the first point on the difficulty of defining tasks, metrics and how to evaluate specific components in Section “More Tasks and Evaluation“. The current benchmarking is focused on perception tasks.

The structure of this retrospective could be more coherent if the authors introduce a more substantial discussion of benchmarking in robotics. We encourage the authors to do so.

Specific comments on paper and terminology:
-  …. to evaluate the effectiveness of the algorithm. -> … to evaluate the effectiveness of an algorithm
- …from using the datasets and benchmarks, this trend …. -> from using datasets and benchmarks. This trend ….
- Definition of motion type: slide and rotate. -> you may want to adopt robotics terminology here and instead of referring to motion, refer to motion constraints or kinematics as described by joint types such as prismatic (you call it sliding) and revolute/rotational (you call it rotate motion).
- … screw cap, we have to adopt a hardcode way. -> how are the other joint or constrain types not hardcoded?

Here are some resources to reflect more deeply on benchmarking in robotics:
What can robotics research learn from computer vision research? https://arxiv.org/abs/2001.02366
Robotics Debates on Benchmarks: https://youtu.be/Mhj6DBw7MYA?t=5376
YCB Benchmarks – Object and Model Set: https://www.ycbbenchmarks.com/
https://robohub.org/robotic-research-are-we-applying-the-scientific-method/

---

### Decision · Program_Chairs · 2020-06-25

Accept